# ADDITIVE POISSON PROCESS: LEARNING INTENSITY OF HIGHER-ORDER INTERACTION IN POINT PROCESSES

## ABSTRACT

We present the *Additive Poisson Process* (APP), a novel framework that can model the higher-order interaction effects of the intensity functions in point processes using lower dimensional projections. Our model combines the techniques in *information geometry* to model higher-order interactions on a statistical manifold and in *generalized additive models* to use lower-dimensional projections to overcome the effects from the curse of dimensionality. Our approach solves a convex optimization problem by minimizing the KL divergence from a sample distribution in lower dimensional projections to the distribution modeled by an intensity function in the point process. Our empirical results show that our model is able to use samples observed in the lower dimensional space to estimate the higher-order intensity function with extremely sparse observations.

## 1 INTRODUCTION

Consider two point processes which are correlated with arrival times for an event. For a given time interval, what is the probability of observing an event from both processes? Can we learn the joint intensity function by just using the observations from each individual processes? Our proposed model, the *Additive Poisson Process* (APP), provides a novel solution to this problem.

The Poisson process is a counting process used in a wide range of disciplines such as time-space sequence data including transportation (Zhou et al., 2018), finance (Ilalan, 2016), ecology (Thompson, 1955), and violent crime (Taddy, 2010) to model the arrival times for a single system by learning an intensity function. For a given time interval of the intensity function, it represents the probability of a point being excited at a given time. Despite the recent advances of modeling of the Poisson processes and its wide applicability, majority of the point processes model do not consider the correlation between two or more point processes. Our proposed approach learns the joint intensity function of the point process which is defined to be the simultaneous occurrence of two events.

For example in a spatial-temporal problem, we want to learn the intensity function for a taxi to pick up customers at a given time and location. For this problem, each point is multi-dimensional, that is $(x, y, t)_{i=1}^{N}$, where a pair of $x$ and $y$ represents two spatial dimensions and $t$ represents the time dimension. For any given location or time, we can only expect very few pick-up events occurring, therefore making it difficult for any model to learn the low valued intensity function.

Previous approaches such as Kernel density estimation (KDE) (Rosenblatt, 1956) are able to learn the joint intensity function. However, KDE suffers from the curse of dimensionality, which means that KDE requires a large size sample or a high intensity function to build an accurate model. In addition, the complexity of the model expands exponentially with respect to the number of dimensions, which makes it infeasible to compute. Bayesian approaches such as using a mixture of beta distributions with a Dirichlet prior (Kottas, 2006) and Reproducing Kernel Hilbert Space (RKHS) (Flaxman et al., 2017) have been proposed to quantify the uncertainty with a prior for the intensity function. However, these approaches are often non-convex, making it difficult to obtain the global optimal solution. In addition, if observations are sparse, it is hard for these approaches to learn a reasonable intensity function.

All previous models are unable to efficiently and accurately learn the intensity of the interaction between point processes. This is because the intensity of the joint process is often low, leading to sparse samples or, in an extreme case, no direct observations of the simultaneous event at all, making it difficult to learn the intensity function from the joint samples. In this paper, we propose a novel framework to learn the higher-order interaction effects of intensity functions in point processes. Our model combines the techniques introduced by Luo & Sugiyama (2019) to model higher-order interactions between point processes and by Friedman & Stuetzle (1981) in *generalized additive model*s to learn the intensity function using samples in a lower dimensional space. Our proposed approach is to decompose a multi-dimensional point process into lower-dimensional representations. For example, in the $x$-dimension we have points $(x_i)_{i=1}^N$, in the $y$-dimension, we have points $(y_i)_{i=1}^N$ and in the time dimension we have $(t_i)_{i=1}^N$. The data in these lower dimensional space can be used to improve the estimate of the joint intensity function. This is different from the traditional approach where we only use the simultaneous events to learn the joint intensity function.

We first show the connection between generalized additive models and Poisson processes. We then provide the connection between generalized additive models and the *log-linear model* (Agresti, 2012), which has a well-established theoretical background in information geometry (Amari, 2016). We draw parallels between the formulation of the generalized additive models and the binary log-linear model on a partially ordered set (poset) (Sugiyama et al., 2017). The learning process in our model is formulated as a convex optimization problem to arrive at a unique optimal solution using natural gradient, which minimizes the Kullback-Leibler (KL) divergence from the sample distribution in a lower dimensional space to the distribution modeled by the learned intensity function. This connection provides remarkable properties to our model: the ability to learn higher-order intensity functions using lower dimensional projections, thanks to the *Kolmogorov-Arnold representation theorem*. This property makes it advantageous to use our proposed approach for the cases where there are, no observations, missing samples, or low event rate. Our model is flexible because it can capture interaction between processes as a partial order structure in the log-linear model and the parameters of the model are fully customizable to meet the requirements for the application. Our empirical results show that our model effectively uses samples projected onto a lower dimensional space to estimate the higher-order intensity function. Our model is also robust to various sample sizes.

## 2 FORMULATION

In this section we first introduce the technical background in the Poisson process and its extension to a multi-dimensional Poisson process. We then introduce the Generalized Additive Model (GAM) and its connection to the Poisson process. This is followed by presenting our novel framework, called Additive Poisson Process (APP), which is our main technical contribution and has a tight link to the Poisson process modelled by GAMs. We show that learning of APP can be achieved via convex optimization using natural gradient.

The Poisson process is characterized by an intensity function $\lambda:\mathbb{R}^D \to \mathbb{R}$, where we assume multiple $D$ processes. An inhomogeneous Poisson process is a general type of processes, where the arrival intensity changes with time. The process with time-changing intensity $\lambda(t)$ is defined as a counting process $\mathbb{N}(t)$, which has an independent increment property. For all time $t \geq 0$ and changes in time $\delta \geq 0$, the probability $p$ for the observations is given as $p(\mathbb{N}(t + \delta) - \mathbb{N}(t) = 0) = 1 - \delta\lambda(t) + o(\delta)$, $p(\mathbb{N}(t + \delta) - \mathbb{N}(t) = 1) = \delta\lambda(t) + o(\delta)$, and $p(\mathbb{N}(t + \delta) - \mathbb{N}(t) \geq 2) = o(\delta)$, where $o(\cdot)$ denotes little-o notation (Daley & Vere-Jones, 2007). Given a realization of timestamps $\mathbf{t}_1, \mathbf{t}_2, \ldots, \mathbf{t}_N$ with $\mathbf{t}_i \in [0, T]^D$ from an inhomogeneous (multi-dimensional) Poisson process with the intensity $\lambda$. Each $\mathbf{t}_i$ is the time of occurrence for the $i$-th event across $D$ processes and $T$ is the observation duration. The likelihood for the Poisson process (Daley & Vere-Jones, 2007) is given by

$$p\left(\{\mathbf{t}_i\}_{i=1}^N \mid \lambda(\mathbf{t})\right) = \exp\left(-\int \lambda(\mathbf{t})\,d\mathbf{t}\right)\prod_{i=1}^N \lambda(\mathbf{t}_i), \qquad (1)$$

where $\mathbf{t} = [t^{(1)}, \ldots, t^{(D)}] \in \mathbb{R}^D$. We define the functional prior on $\lambda(\mathbf{t})$ as

$$\lambda(\mathbf{t}) := g(f(\mathbf{t})) = \exp(f(\mathbf{t})). \qquad (2)$$

The function $g(\cdot)$ is a positive function to guarantee the non-negativity of the intensity which we choose to be the exponential function, and our objective is to learn the function $f(\cdot)$. The log-

likelihood of the multi-dimensional Poisson process with the functional prior is described as

$$\log p\left(\{\mathbf{t}_i\}_{i=1}^N \mid \lambda(\mathbf{t})\right) = \sum_{i=1}^N f(\mathbf{t}_i) - \int \exp(f(\mathbf{t}))\,d\mathbf{t}. \tag{3}$$

In the following sections, we introduce *generalized additive models* and propose to model it by the *log-linear model* to learn $f(\mathbf{t})$ and the normalizing term.

## 2.1 GENERALIZED ADDITIVE MODEL

In this section we present the connection between Poisson processes with Generalized Additive Model (GAM) proposed by Friedman & Stuetzle (1981). GAM projects higher-dimensional features into lower-dimensional space to apply smoothing functions to build a restricted class of non-parametric regression models. GAM is less affected by the curse of dimensionality compared to directly using smoothing in a higher-dimensional space. For a given set of processes $J \subseteq [D] = \{1, \ldots, D\}$, the traditional GAM using one-dimensional projections is defined as $\log \lambda_J(\mathbf{t}) = \sum_{j \in J} f_j(t^{(j)}) - \beta_J$ with some smoothing function $f_j$.

In this paper, we extend it to include higher-order interactions between features in GAM. The *k-th order GAM* is defined as

$$\log \lambda_J(\mathbf{t}) = \sum_{j \in J} f_{\{j\}}(t^{(j)}) + \sum_{j_1, j_2 \in J} f_{\{j_1, j_2\}}(t^{(j_1)}, t^{(j_2)}) + \cdots + \sum_{j_1, \ldots, j_k \in J} f_{\{j_1, \ldots, j_k\}}(t^{(j_1)}, \ldots, t^{(j_k)}) - \beta_J$$

$$= \sum_{I \subseteq J, |I| \leq k} f_I(\mathbf{t}^{(I)}) - \beta_J, \tag{4}$$

where $\mathbf{t}^{(I)} \in \mathbb{R}^{|I|}$ denotes the subvector $(\mathbf{t}^{(j)})_{j \in I}$ of $\mathbf{t}$ with respect to $I \subseteq [D]$. The function $f_I : \mathbb{R}^{|I|} \to \mathbb{R}$ is a smoothing function to fit the data, and the normalization constant $\beta_J$ for the intensity function is obtained as $\beta_J = \int \lambda_J(\mathbf{t})d\mathbf{t} = \int \exp(\sum_{I \subseteq J, |I| \leq k} f_I(\mathbf{t}^{(I)}))d\mathbf{t}$. The definition of the additive model is in the same form as Equation (3). In particular, if we compare Equation (3) and (4), we can see that the smoothing function $f$ in (3) corresponds to the right-hand side of (4).

Learning of a continuous function using lower dimensional projections is well known because of the *Kolmogorov-Arnold representation theorem*, which states as follows:

**Theorem 1** (Kolmogorov–Arnold Representation Theorem (Braun & Griebel, 2009; Kolmogorov, 1957)). *Any multivariate continuous function can be represented as a superposition of one–dimensional functions, i.e.,* $f(t_1, \ldots, t_n) = \sum_{q=1}^{2n+1} f_q\left(\sum_{p=1}^n g_{q,p}(t_p)\right)$.

Braun (2009) showed that the GAM is an approximation to the general form presented in Kolmogorov-Arnold representation theorem by replacing the range $q \in \{1, \ldots, 2n+1\}$ with $I \subseteq J$ and the inner function $g_{q,p}$ by the identity if $q = p$ and zero otherwise, yielding $f(\mathbf{t}) = \sum_{I \subseteq J} f_I(\mathbf{t}^{(I)})$.

Interestingly, the canonical form for additive models in Equation (4) can be rearranged to be in the same form as Kolmogorov-Arnold representation theorem. By letting $f(\mathbf{t}) = \sum_{I \subseteq J} f_I(\mathbf{t}^{(I)}) = g^{-1}(\lambda(\mathbf{t}))$ and $g(\cdot) = \exp(\cdot)$, we have

$$\lambda_J(\mathbf{t}) = \frac{1}{\exp(\beta_J)} \exp\left(\sum_{I \subseteq J} f_I\left(\mathbf{t}^{(I)}\right)\right) \propto \exp\left(\sum_{I \subseteq J} f_I\left(\mathbf{t}^{(I)}\right)\right), \tag{5}$$

where we assume $f_I(\mathbf{t}^{(I)}) = 0$ if $|I| > k$ for the $k$-th order model and $1/\exp(\beta_J)$ is the normalization term for the intensity function. Based on the Kolmogorov-Arnold representation theorem, generalized additive models are able to learn the intensity of the higher-order interaction between point processes by using projections into lower dimensional space. The log-likelihood function for a $k$th-order model is obtained by substituting the Equation (4) into Equation (1),

$$\log p\left(\{\mathbf{t}\}_{i=1}^N \mid \lambda(\mathbf{t})\right) = \sum_{i=1}^N \exp\left(\sum_{I \subseteq J, |I| \leq k} f_I\left(\mathbf{t}^{(I)}\right)\right) - \beta',$$

where is a constant given by $\beta' = \int \lambda(\mathbf{t})d\mathbf{t} + \sum_{I \subseteq J} \beta_J$. In the following section we will detail a log-linear formulation that efficiently maximizes this log-likelihood equation.

## 2.2 ADDITIVE POISSON PROCESS

We introduce our key technical contribution in this section, the log-linear formulation of the *additive Poisson process*, and draw parallels between higher-order interactions in the log-linear model and the lower dimensional projections in generalized additive models. In the following, we discretize the time window $[0, T]$ into $M$ bins and treat each bin as a natural number $\tau \in [M] = \{1, 2, \ldots, M\}$ for each process. We assume that $M$ is predetermined by the user. First we introduce a structured space for the Poisson process to incorporate interactions between processes. Let $\Omega = \{(J, \tau) | J \in 2^{[D]} \setminus \emptyset, \tau \in [M]\} \cup \{(\perp, 0)\}$. We define the *partial order* $\preceq$ (Davey & Priestley, 2002) on $\Omega$ as

$$(J, \tau) \preceq (J', \tau') \iff J \subseteq J' \text{ and } \tau \le \tau', \quad \text{for each } \omega = (J, \tau), \omega' = (J', \tau') \in \Omega, \quad (6)$$

and $(\perp, 0) \preceq (J, \tau)$ for all $(J, \tau) \in \Omega$, which is illustrated in Figure 1. The relation $J \subseteq J'$ is used to model any-order interactions between point processes (Luo & Sugiyama, 2019) (Amari, 2016, Section 6.8.4) and each $\tau$ in $(J, \tau)$ represents "time" in our model with $\perp$ denoting the least element in the partial order structure. Note that the domain of $\tau$ can be generalized from $[M]$ to $[M]^D$ to take different time stamps into account, while in the following we assume that observed time stamps are always the same across processes for simplic-

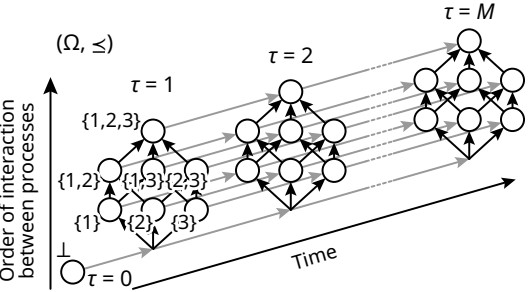

Figure 1: Partial order structured sample space $(\Omega, \preceq)$ with $D = 3$.

ity. Our experiments in the next section demonstrates that we can still accurately estimate the density of processes. Our model can be applied to not only time-series data but any sequential data.

On any set equipped with a partial order, we can introduce a *log-linear model* (Sugiyama et al., 2016; 2017). Given a parameter domain $\mathcal{S} \subseteq \Omega$. For a partially ordered set $(\Omega, \preceq)$, the log-linear model with parameters $(\theta_s)_{s \in \mathcal{S}}$ is introduced as

$$\log p(\omega; \theta) = \sum_{s \in \mathcal{S}} \mathbf{1}_{[s \preceq \omega]} \theta_s - \psi(\theta) \quad (7)$$

for each $\omega \in \Omega$, where $\mathbf{1}_{[\cdot]} = 1$ if the statement in $[\cdot]$ is true and 0 otherwise, and $\psi(\theta) \in \mathbb{R}$ is the partition function uniquely obtained as $\psi(\theta) = \log \sum_{\omega \in \Omega} \exp(\sum_{s \in \mathcal{S}} \mathbf{1}_{[s \preceq \omega]} \theta_s) = -\theta_{(\perp, 0)}$. A special case of this formulation coincides with the density function of the *Boltzmann machines* (Sugiyama et al., 2018; Luo & Sugiyama, 2019).

Here we have a clear correspondence between the log-linear formulation and that in the form of Kolmogorov-Arnold representation theorem in Equation (5) if we rewrite Equation (7) as

$$p(\omega; \theta) = \frac{1}{\exp \psi(\theta)} \exp \left( \sum_{s \in \mathcal{S}} \mathbf{1}_{[s \preceq \omega]} \theta_s \right) \propto \exp \left( \sum_{s \in \mathcal{S}} \mathbf{1}_{[s \preceq \omega]} \theta_s \right). \quad (8)$$

We call this model with $(\Omega, \preceq)$ defined in Equation (6) the additive Poisson process, which represents the intensity $\lambda$ as the joint distribution across all possible states. The intensity $\lambda$ of the multi-dimensional Poisson process given via the GAM in Equation (5) is fully modeled (parameterized) by Equation (7) and each intensity $f_I(\cdot)$ is obtained as $\theta_{(I, \cdot)}$. To consider the $k$-th order model, we consistently use the parameter domain $\mathcal{S}$ given as $\mathcal{S} = \{(J, \tau) \in \Omega \mid |J| \le k\}$, where $k$ is an input parameter to the model that specifies the upper bound of the order of interactions. This means that $\theta_s = 0$ for all $s \notin \mathcal{S}$. Note that our model is well-defined for any subset $\mathcal{S} \subseteq \Omega$ and the user can use arbitrary domain in applications.

For a given $J$ and each bin $\tau$ with $\omega = (J, \tau)$, the empirical probability $\hat{p}(\omega)$, which corresponds to the input observation, is given as

$$\hat{p}(\omega) = \frac{1}{Z} \sum_{I \subseteq J} \sigma_I(\boldsymbol{\tau}), \quad Z = \sum_{\omega \in \Omega} \hat{p}(\omega), \quad \text{and} \quad \sigma_I(\boldsymbol{\tau}) := \frac{1}{N h_I} \sum_{i=1}^{N} K \left( \frac{\boldsymbol{\tau}^{(I)} - \mathbf{t}_i^{(I)}}{h_I} \right) \quad (9)$$

for each discretized state $\omega = (J, \tau)$, where $\boldsymbol{\tau} = (\tau, \ldots, \tau) \in \mathbb{R}^D$. The function $\sigma_I$ performs smoothing on time stamps $\mathbf{t}_1, \ldots, \mathbf{t}_N$, which is the kernel smoother proposed by Buja et al. (1989). The function $K$ is a kernel and $h_I$ is the bandwidth for each projection $I \subseteq [D]$. We use the Gaussian kernel as $K$ to ensure that probability is always nonzero, meaning that the definition of the kernel smoother coincides with the kernel estimator of the intensity function proposed by Schäbe (1993).

## 2.3 OPTIMIZATION

Given an empirical distribution $\hat{p}$ defined in Equation (9), the task is to learn the parameter $(\theta_s)_{s \in \mathcal{S}}$ such that the distribution via the log-linear model in Equation (7) is close to $\hat{p}$ as much as possible. Let us define $\mathfrak{S}_{\mathcal{S}} = \{p \mid \theta_s = 0 \text{ if } s \notin \mathcal{S}\}$, which is the set of distributions that can be represented by the log-linear model using the parameter domain $\mathcal{S}$. Then the objective function is given as $\min_{p \in \mathfrak{S}_{\mathcal{S}}} D_{\mathrm{KL}}(\hat{p}, p)$, where $D_{\mathrm{KL}}(\hat{p}, p) = \sum_{\omega \in \Omega} \hat{p} \log(\hat{p}/p)$ is the KL divergence from $\hat{p}$ to $p$. In this optimization, let $p^*$ be the learned distribution from the sample with infinitely large sample size and $p$ be the learned distribution for each sample. Then we can lower bound the uncertainty (variance) $\mathbb{E}[D_{\mathrm{KL}}(p^*, p)]$ by $|\mathcal{S}|/2N$ (Barron & Hengartner, 1998).

Thanks to the well developed theory of *information geometry* (Amari, 2016) for the log-linear model (Amari, 2001), it is known that this problem can be solved by *e-projection*, which coincides with the maximum likelihood estimation, and it is always *convex optimization* (Amari, 2016, Chapter 2.8.3). The gradient with respect to each parameter $\theta_s$ is obtained by $(\partial/\partial\theta_s)D_{\mathrm{KL}}(\hat{p}, p) = \eta_s - \hat{\eta}_s$, where $\eta_s = \sum_{\omega \in \Omega} \mathbf{1}_{[\omega \succeq s]} p(\omega)$. The value $\eta_s$ is known as the expectation parameter (Sugiyama et al., 2017) and $\hat{\eta}_s$ is obtained by replacing $p$ with $\hat{p}$ in the above equation. If $\hat{\eta}_s = 0$ for some $s \in \mathcal{S}$, we remove $s$ from $\mathcal{S}$ to ensure that the model is well-defined.

Let $\mathcal{S} = \{s_1, \ldots, s_{|\mathcal{S}|}\}$ and $\boldsymbol{\theta} = [\theta_{s_1}, \ldots, \theta_{s_{|\mathcal{S}|}}]^T$, $\boldsymbol{\eta} = [\eta_{s_1}, \ldots, \eta_{s_{|\mathcal{S}|}}]^T$. We can always use the *natural gradient* (Amari, 1998) as the closed form solution of the Fisher information matrix is always available (Sugiyama et al., 2017). The update step is $\boldsymbol{\theta}_{\text{next}} = \boldsymbol{\theta} - \mathbf{G}^{-1}(\boldsymbol{\eta} - \hat{\boldsymbol{\eta}})$, where the Fisher information matrix $\mathbf{G}$ is obtained as

$$g_{ij} = \frac{\partial}{\partial\theta_{s_i}\partial\theta_{s_j}} D_{\mathrm{KL}}(\hat{p}, p) = \sum_{\omega \in \Omega} \mathbf{1}_{[\omega \succeq s_i]}\mathbf{1}_{[\omega \succeq s_j]} p(\omega) - \eta_{s_i}\eta_{s_j}. \tag{10}$$

Theoretically the Fisher information matrix is numerically stable to perform a matrix inversion. However, computationally floating point errors may cause the matrix to become indefinite. To overcome this issue, a small positive value is added along the main diagonal of the matrix. This technique is known as jitter and it is used in areas like Gaussian processes to ensure that the covariance matrix is computationally positive semi-definite (Neal, 1999).

The pseudocode for APP is shown in Algorithm 1. The time complexity of computing line 7 is $\mathcal{O}(|\Omega||\mathcal{S}|)$. This means when implementing the model using gradient descent, the time complexity of the model is $\mathcal{O}(|\Omega||\mathcal{S}|^2)$ to update the parameters in $\mathcal{S}$ for each iteration. For natural gradient the cost of inverting the Fisher information matrix $G$ is $\mathcal{O}(|\mathcal{S}|^3)$, therefore the time complexity to update the parameters in $\mathcal{S}$ is

---

**Algorithm 1** Additive Poisson Process (APP)

1: **Function** APP$(\{\mathbf{t}_i\}_{i=1}^N, \mathcal{S}, M, h)$:
2: Initialize $\Omega$ with the number $M$ of bins
3: Apply Gaussian Kernel with bandwidth $h$ on $\{\mathbf{t}_i\}_{i=1}^N$ to compute $\hat{p}$
4: Compute $\hat{\boldsymbol{\eta}} = (\hat{\eta}_s)_{s \in \mathcal{S}}$ from $\hat{p}$
5: Initialize $\boldsymbol{\theta} = (\theta_s)_{s \in \mathcal{S}}$ (randomly or $\theta_s = 0$)

6: **repeat**
7:     Compute $p$ using the current $\boldsymbol{\theta} = (\theta_s)_{s \in \mathcal{S}}$
8:     Compute $\boldsymbol{\eta} = (\eta_s)_{s \in \mathcal{S}}$ from $p$
9:     $\Delta\boldsymbol{\eta} \leftarrow \boldsymbol{\eta} - \hat{\boldsymbol{\eta}}$
10:     Compute the Fisher information matrix $\mathbf{G}$ using Equation (10)
11:     $\boldsymbol{\theta} \leftarrow \boldsymbol{\theta} - \mathbf{G}^{-1}\Delta\boldsymbol{\eta}$
12: **until** convergence of $\boldsymbol{\theta} = (\theta_s)_{s \in \mathcal{S}}$
13: **End Function**

---

$\mathcal{O}(|\mathcal{S}|^3 + |\Omega||\mathcal{S}|)$ for each iteration. The time complexity for natural gradient is significantly higher to invert the fisher information matrix, if the number of parameter is small, it is more efficient to use natural gradient because it requires significantly less iterations. However, if the number of parameters is large, it is more efficient to use gradient descent.

## 3 EXPERIMENTS

We perform experiments using two dimensional synthetic data, higher dimensional synthetic data, and rea-world data to evaluate the performance of our proposed approach. Our code is implemented on Python 3.7.5 with NumPy version 1.8.2 and the experiments are run on Ubuntu 18.04 LTS with an Intel i7-8700 6c/12t with 16GB of memory [1]. In experiments of synthetic data, we simulate

---

[1]The code is available in the supplementary material and will be publicly available online after the peer review process.

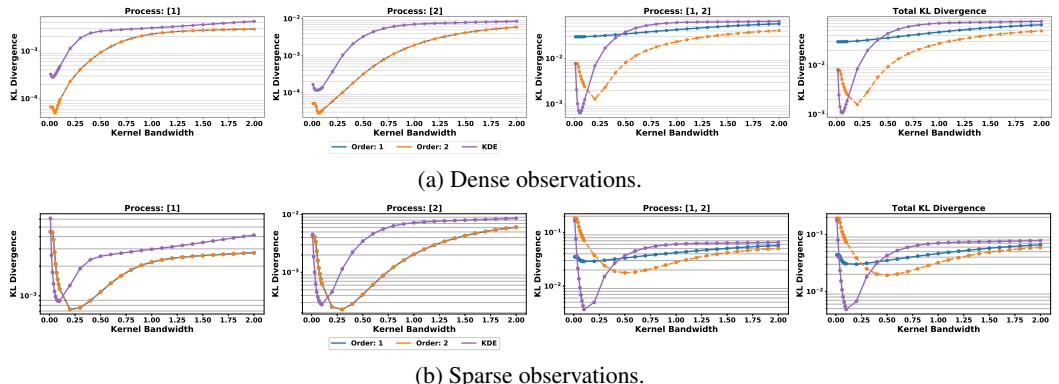

(a) Dense observations.

(b) Sparse observations.

Figure 2: KL Divergence for four-order Poisson process.

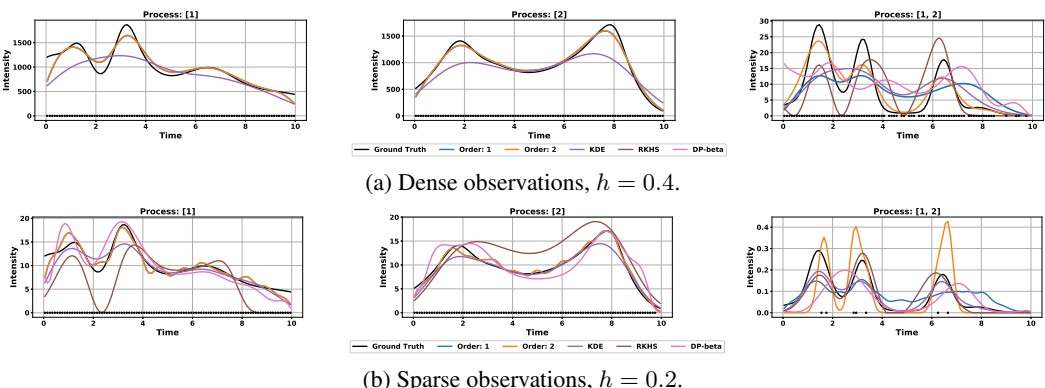

(a) Dense observations, $h = 0.4$.

(b) Sparse observations, $h = 0.2$.

Figure 3: Intensity function of two dimensional processes. Dots represent observations.

random events using Equation (1). We generate an intensity function using a mixture of Gaussians, where the mean is drawn from a uniform distribution and the covariance is drawn from an inverted Wishart distribution. The intensity function is then the density function multiplied by the sample size. The synthetic data is generated by directly drawing a sample from the probability density function . An arbitrary number of samples is drawn from the mixture of Gaussians. We then run our models and compare with Kernel Density Estimation (KDE) (Rosenblatt, 1956), an inhomogeneous Poisson process whose intensity is estimated by a reproducing kernel Hilbert space formulation (RKHS) (Flaxman et al., 2017), and a Dirichlet process mixture of Beta distributions (DP-beta) (Kottas, 2006). The hyper-parameters $M$ and $h$ in our proposed model are selected using grid search and cross-validation. For situations where a validation set is not available, then $h$ could be selected using a rule of thumb approach such as Scott's Rule (Scott, 2015) and $M$ could be selected empirically from the input data by computing the time interval of the joint observation.

## 3.1 EXPERIMENTS ON TWO-DIMENSIONAL PROCESSES

For our experiment, we use 20 Gaussian components and simulate a dense case with 100,000 observations and a sparse case with 1,000 observations within the time frame of 10 seconds. We consider that a joint event occurs if the two events occur 0.1 seconds apart. Figure 2a and Figure 2b compares the KL divergence between the first- and second-order models and Figure 3 are the corresponding intensity functions. In the first-order processes, both first- and second-order models have the same performance. This is expected as both of the model can treat first-order interactions and is able to learn the empirical intensity function exactly which is the superposition of the one-dimensional projection of the Gaussian kernels on each observation. For the second-order process, the second-order model performs better than the first-order model because it is able to directly learn the intensity function from the projection onto the two-dimensional space. In contrast, the first-order model must approximate the second-order process using the observations from the first order-processes. In the sparse case, the second-order model performs better when the correct bandwidth is selected.

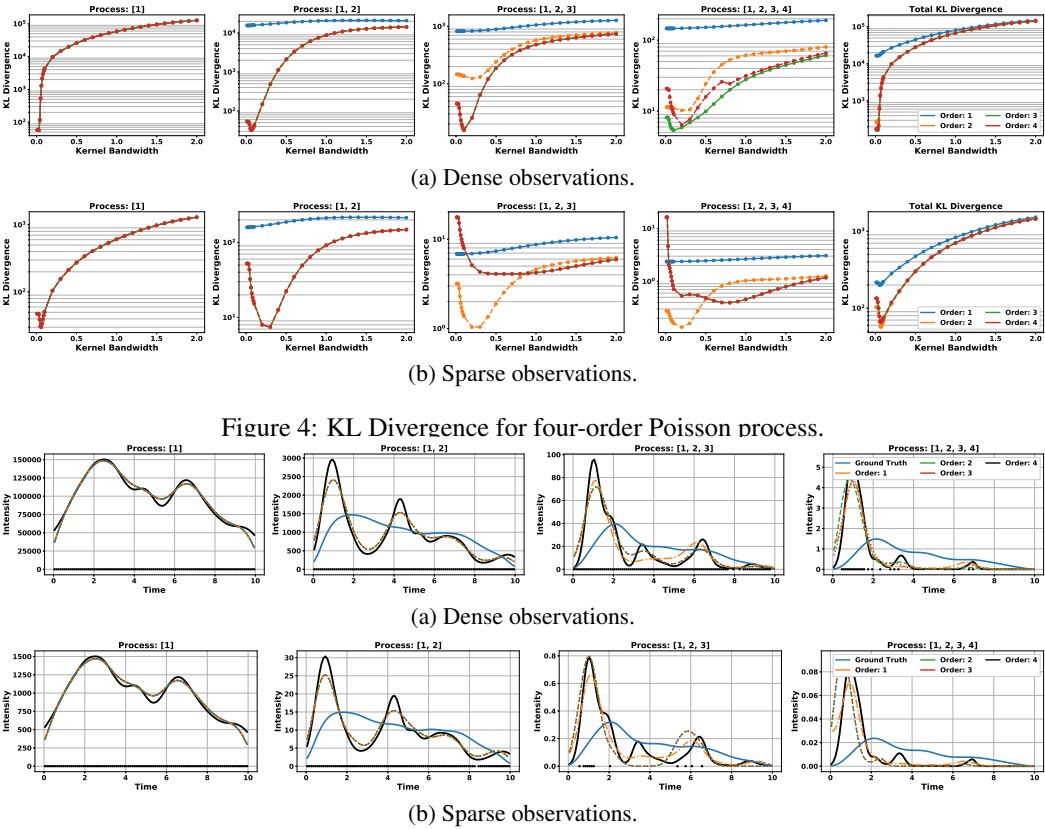

(a) Dense observations.

(b) Sparse observations.

Figure 4: KL Divergence for four-order Poisson process.

(a) Dense observations.

(b) Sparse observations.

Figure 5: Intensity function of higher dimensional processes. Dots represent observations.

Table 1 compares our approach APP with other state-of-the-art approaches. APP performs the best for first-order processes in both the sparse and dense experiments. Experiments for RKHS and DP-beta were unable to complete running within 2 days for the dense experiment. In the second-order process our approach was outperformed by KDE, while both the second-order APP is able to outperform both RKHS and DP-beta process for both sparse and dense experiments. Figure 2a and Figure 2b show that KDE is sensitive to changes in bandwidth, which means that, for any practical implementation of the model, second-order APP with a less sensitive bandwidth is more likely to learn a more accurate intensity function when the ground truth is unknown.

## 3.2 Experiments on Higher-Dimensional Processes

We generate a fourth-order process to simulate the behaviour of the model in higher dimensions. The model is generalizable to higher dimensions, however it is difficult to demonstrate results for processes higher than fourth-order. For our experiment, we generate an intensity function using 50 Gaussian components and draw a sample with the size of $10^7$ for the dense case and that with the size of $10^5$ for the sparse case. We consider the joint event to be the time frame of 0.1 seconds.

We were not able to run comparison experiments with other models because they are unable to learn when there are no or few direct observations in third- and fourth-order processes. In addition, the time complexity is too high to learn from direct observations in first- and second-order processes because all the other models have their time complexity proportional to the number of observations. The time complexity for KDE is $\mathcal{O}(N^D)$ for the dimensionality with $D$, while DP-beta is $\mathcal{O}(N^2K)$, where $K$ is the number of clusters, and RKHS is $\mathcal{O}(N^2)$ for each iteration with respect to the sample size $N$, where DP-beta and RKHS are applied to a single dimension as they cannot directly treat multiple dimensions. KDE is able to make an estimation of the intensity function when there are no direct observations of the simultaneous event, however, it was too computationally expensive to complete running the experiment. Differently, our model is more efficient because the time complexity is proportional to the number of bins in our model. The time complexity of APP for each iteration is $\mathcal{O}(|\Omega||\mathcal{S}|)$, where $|\Omega| = M^D$ and $|\mathcal{S}| = \sum_{c=1}^{k} \binom{D}{c}$. Our model scales combinatorially

Table 1: The lowest KL divergence from the ground truth distribution to the obtained distribution on two types of single processes ([1] and [2]) and joint process of them ([1,2]). APP-# represents the order of the Additive Poisson Process. Missing values mean that the computation did not finish within two days.

| | Process | APP-1 | APP-2 | KDE | RKHS | DP-beta |
|---|---|---|---|---|---|---|
| Dense | [1] | **4.98e-5** | **4.98e-5** | 2.81e-4 | - | - |
| | [2] | **2.83e-5** | **2.83e-5** | 1.17e-4 | - | - |
| | [1,2] | 2.98e-2 | 1.27e-3 | **6.33e-4** | 4.09e-2 | 4.54e-2 |
| Sparse | [1] | **7.26e-4** | **7.26e-4** | 8.83e-4 | 1.96e-2 | 2.62e-3 |
| | [2] | **2.28e-4** | **2.28e-4** | 2.76e-4 | 2.35e-3 | 2.49e-3 |
| | [1,2] | 2.88e-2 | 1.77e-2 | **3.67e-3** | 1.84e-2 | 3.68e-2 |

Table 2: Negative test log-likelihood for the New York Taxi data. Single processes ([T] and [W]) and joint process of them ([T,W]). APP-# represents the order of the Additive Poisson Process.

| | Process | APP-1 | APP-2 | KDE | RKHS | DP-beta |
|---|---|---|---|---|---|---|
| Jan | [T] | 714.07 | 714.07 | **713.77** | 728.13 | 731.01 |
| | [W] | 745.60 | 745.60 | **745.23** | 853.42 | 790.04 |
| | [T,W] | 249.60 | **246.05** | 380.22 | 259.29 | 260.30 |
| Feb | [T] | **713.43** | **713.43** | 755.71 | 795.61 | 765.76 |
| | [W] | **738.66** | **738.66** | 773.65 | 811.34 | 792.10 |
| | [T,W] | 328.84 | **244.21** | 307.86 | 334.31 | 326.52 |
| Mar | [T] | **716.72** | **716.72** | 733.74 | 755.48 | 741.28 |
| | [W] | **738.06** | **738.06** | 816.99 | 853.33 | 832.43 |
| | [T,W] | 291.20 | **246.19** | 289.69 | 328.47 | 300.36 |

with respect to the number of dimensions. However, this is unavoidable for any model which directly takes into account the high-order interactions. For practical applications, the number of dimensions $D$ and the order of the model $k$ is often small, making it feasible to compute.

In Figure 4a we observe similar behaviour in the model, where the first-order processes fit precisely to the empirical distribution generated by the Gaussian kernels. The third-order model is able to period better on the fourth-order process. This is because the observation shown in Figure 5a is largely sparse and learning from the observations directly may overfit. A lower dimensional approximation is able to provide a better result in the third-order model. Similar trends can be seen in the sparse case as shown in Figure 4b, where a second-order model is able to produce better estimation in third- and fourth-order processes. The observations are extremely sparse as seen in Figure 5b, where there are only a few observations or no observations at all to learn the intensity function.

### 3.3 UNCOVERING COMMON PATTERNS IN THE NEW YORK TAXI DATASET

We demonstrate the capability of our model on the 2016 Green Taxi Trip dataset[2]. We are interested in finding the common pick up patterns across Tuesday and Wednesday. We define a common pick up time to be within 1 minute intervals of each other between the two days. We have chosen to learn an intensity function using the Poisson process for Tuesday and Wednesday and a joint process for both of them. The joint process uncovers the common pick up patterns between the two days. We have selected to use the first two weeks of Tuesday and Wednesday in January 2016 as our training and validation sets and Tuesday and Wednesday of the third week of January 2016 as our testing set. We repeat the same experiment for February and March.

We show our results in Table 2, where we use the negative test log-likelihood as an evaluation measure. APP-2 has consistently outperformed all the other approaches for the joint process between Tuesday and Wednesday. In addition, for the individual process, APP-1 and -2 also showed the best result for February and March. These results demonstrate the effectiveness of our model in capturing higher-order interactions between processes, which is difficult for the other existing approaches.

## 4 CONCLUSION

We have proposed a novel framework, called *Additive Poisson Process* (APP), to learn the intensity function of the higher-order interaction between point processes using samples from lower dimensional projections. We formulated our proposed model using the *the log-linear model* and optimize it using information geometric structure of the distribution space. We drew parallels between our proposed model and *generalized additive model* and showed the ability to learn from lower dimensional projections via the *Kolmogorov-Arnold representation theorem*. Our empirical results show the superiority of our method when learning the higher-order interactions between point processes when there are no or extremely sparse direct observations, and our model is also robust to varying sample sizes. Our approach provides a novel formulation to learn the joint intensity function which typically has extremely low intensity. There is enormous potential to apply APP to real-world applications, where higher order interaction effects need to be model such as in transportation, finance, ecology, and violent crimes.

---

[2]https://data.cityofnewyork.us/Transportation/2016-Green-Taxi-Trip-Data/hvrh-b6nb

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

## A    ADDITIONAL EXPERIMENTS

### A.1    BANDWIDTH SENSITIVITY ANALYSIS

Our first experiment is to demonstrate the ability for our proposed model to learn an intensity function from samples. We generate a Bernoulli process with probably of $p = 0.1$ to generate samples for every 1 seconds for 100 seconds to create a toy problem for our model. This experiment is to observe the behaviour of varying the bandwidth in our model. In Figure 6a, we observe that applying no kernel, we learn the deltas of each individual observation. When we apply a Gaussian kernel, the output of the model for the intensity function is much more smooth. Increasing the bandwidth of the kernel will provide a wider and much smoother function. Between the 60 seconds and 80 seconds mark, it can be seen when two observations have overlapping kernels, the intensity function becomes larger in magnitude.

### A.2    ONE DIMENSIONAL POISSON PROCESS

A one dimensional experiment is simulated using Ogata's thinning algorithm (Ogata, 1981). We generate two experiments use the standard sinusoidal benchmark intensity function with a frequency of $20\pi$. The dense experiment has troughs with 0 intensity and peaks at 201 and the sparse experiment has troughs with 0 intensity and peaks at 2. Figure 6d shows the experimental results of the dense case, our model has no problem learning the intensity function. We compare our results using KL divergence between the underlying intensity function used to generate the samples to the intensity function generated by the model. Figure 6b shows that the optimal bandwidth is $h = 1$.

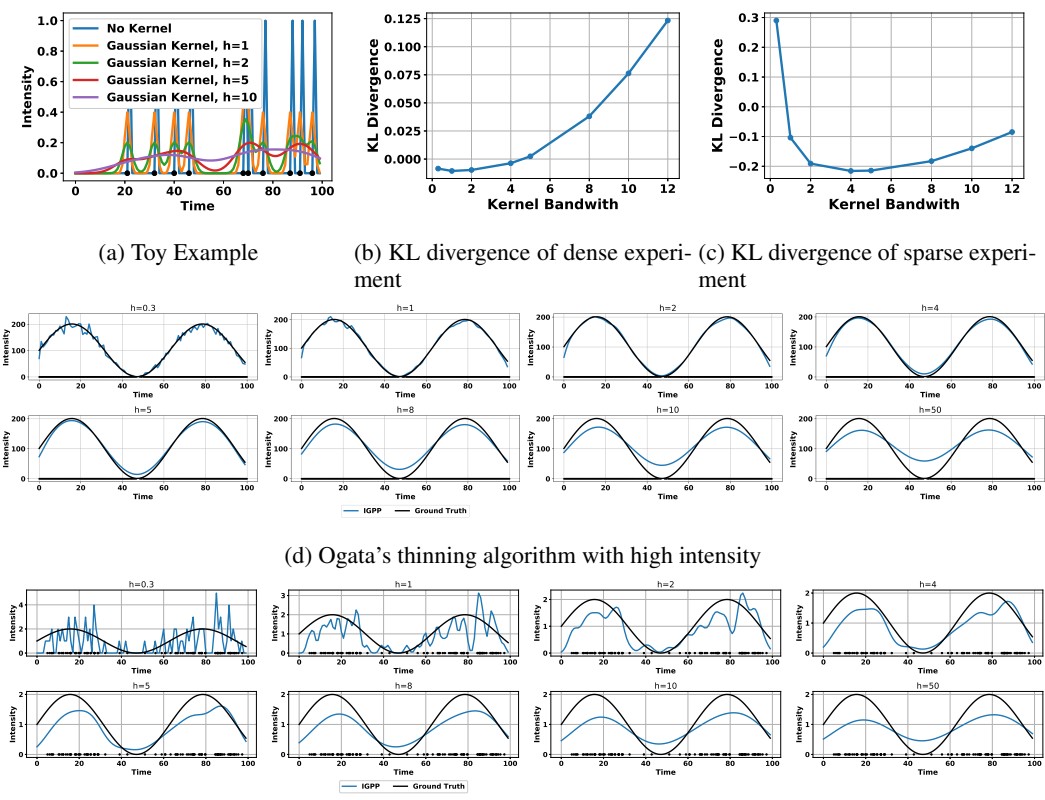

(a) Toy Example

(b) KL divergence of dense experiment

(c) KL divergence of sparse experiment

(d) Ogata's thinning algorithm with high intensity

(e) Ogata's thinning algorithm with low intensity

Figure 6: One dimensional experiments

---

**Algorithm 2** Thinning Algorithm for non-homogenous Poisson Process

1: **Function** Thinning Algorithm $(\lambda(t), T)$:
2: $n = m = 0, t_0 = s_0 = 0, \bar{\lambda} = \sup_{0 \leq t \leq T} \lambda(t)$
3: **repeat**
4:    $u \sim \text{uniform}(0, 1)$
5:    $w = -\frac{1}{\bar{\lambda}} \ln u \{w \sim \text{exponential}(\bar{\lambda})\}$
6:    $s_{m+1} = s_m + w$
7:    $D \sim \text{uniform}(0, 1)$
8:    **if** $D \leq \frac{\lambda(s_{m+1})}{\bar{\lambda}}$ **then**
9:       $t_{n+1} = s_{m+1}$
10:     $n = n + 1$
11:    **else**
12:       $m = m + 1$
13:    **end if**
14:    **if** $t_n \leq T$ **then**
15:       **return** $\{t_k\}_{k=1,2,...,n}$
16:    **else**
17:       **return** $\{t_k\}_{k=1,2,...,n-1}$
18:    **end if**
19: **until** $s_m \leq T$
20: **End Function**

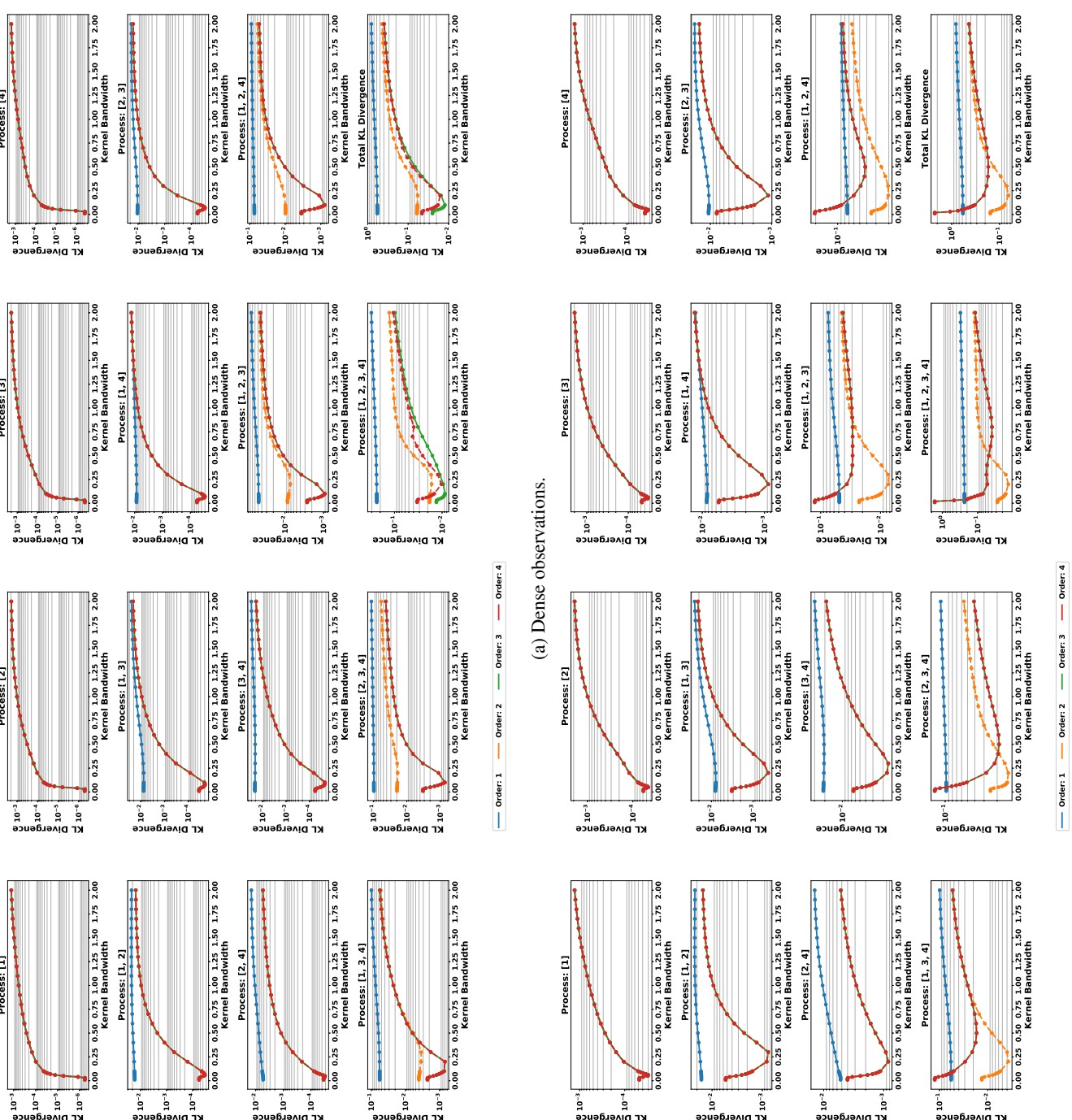

(a) Dense observations.

(b) Sparse observations.

Figure 7: KL Divergence for four-order Poisson process.

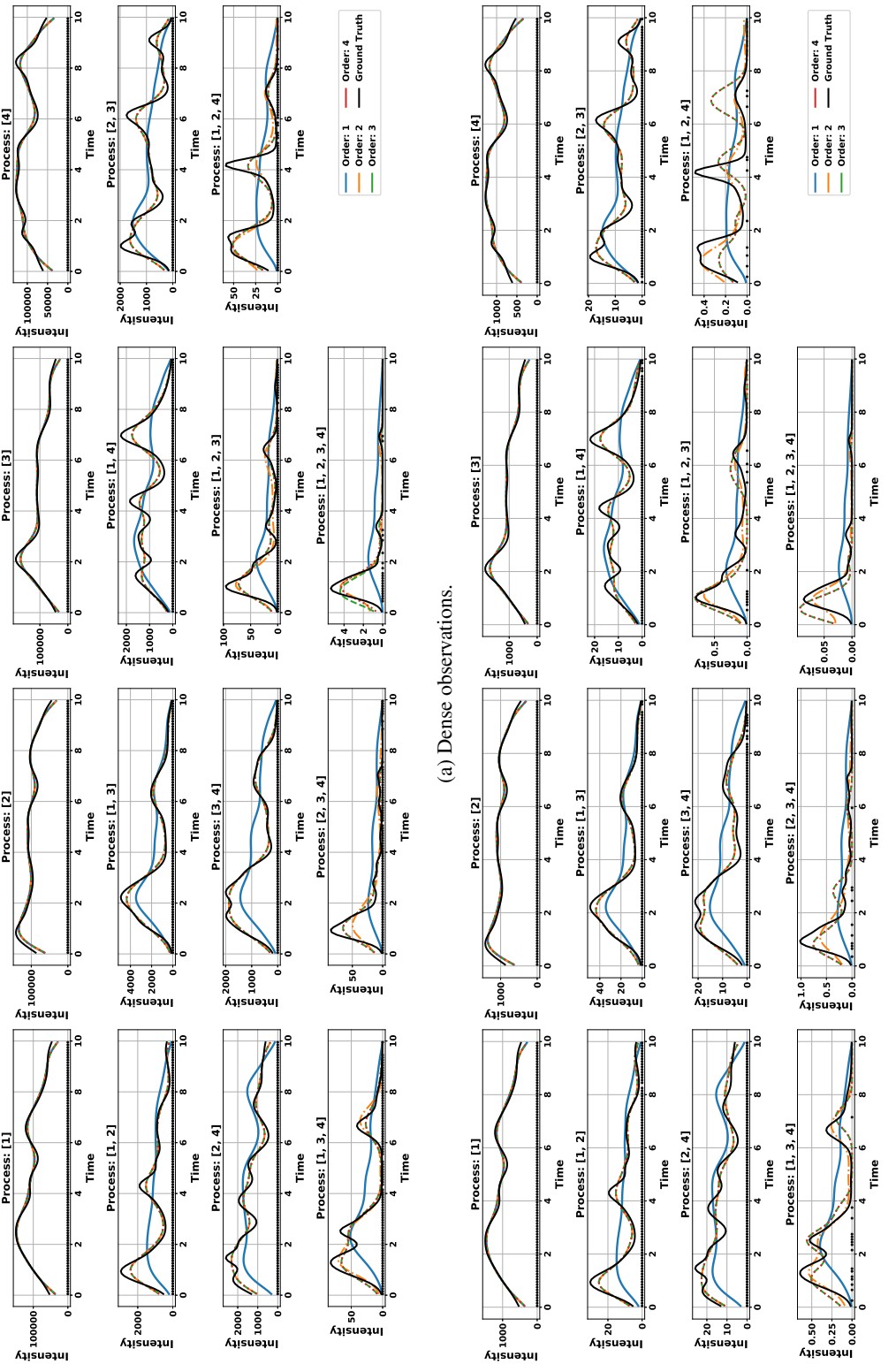

Figure 8: Intensity function of higher dimensional processes. Dots represent observations.

