# OpenReview forum: "Additive Poisson Process: Learning Intensity of Higher-Order Interaction in Stochastic Processes"
_ICLR.cc/2021/Conference — Reject_

### Official Review · AnonReviewer3 · 2020-10-26
**The paper is well written but some part lacks of details making the exposition sometime difficult to follow. Overall the contribution is oroginal and promising, but in its current state suffers from imprecisions making difficult to assess its quality.**

**Rating:** 6
**Confidence:** 3

**Review:**

The paper under review proposes a new model for multi-dimensional temporal Point processes, allowing efficient estimation of high order interactions. This new model, called additive Poisson process, relies on a log-linear structure of the intensity function that is motivated thanks to the Kolmogorov-Arnold theorem. Such structure is then linked to generalized additive models, a result that is used to devise an efficient estimation procedure with formal guarantees  of convergence.

The paper is well written but some part lacks of details making the exposition sometime difficult to follow. Overall the contribution is original and promising, but in its current state suffers from imprecision making difficult to assess its quality. The following points summarizes these remarks:


## Major points

* The experimental design is unclear: what is the structure of the intensity function? Is it log-linear? If yes, why APP, which is designed for this case is outperformed by KDD? Why is the interaction between processes 1 and 2, is summarized by a univariate function ? Does this imply that only simultaneous interactions are considered in this work, as suggested by the second sentence of Section 3.1?
* How would the method compare to others when the log-linearity is not satisfied, e.g., $\lambda(t_1,t_2) = \lambda_1(t_1)\lambda_2(t_2)$? Such cases, if not studied in detail, should at least be mentioned and discussed as a limitation of the current model.
* The application is used to showcase the usefulness of the new model, but model comparison relies on a negative log-likelihood criteria which does not account for the complexity of the models. If possible, some more general criteria such as AIC should be considered. If such calculations are not possible/does not apply, then differences in model's complexity should be at least discussed.

## Minor points

* p.1: the first sentence is really unclear. Not all stochastic process have an 'arrival time'. Note that correlation does not mean simultaneity of the events.
* p.1: 'intensity function is multiplied ...'
* p.1 : twice 'the'.
* p,1 'Differently' sounds awkward.
* p.1: 'if observations are sparse'.
* p.2: 'intensity is often low' would be better to say 'lower than lower-orders'.
* p.2: of the model 'are'.
* p.2: 'the model is robust to' this is not clear and never discussed in the paper. No misspecified experiment is presented.
* p.2 'arrival intensity'?
* p.2: 'for time t > 0, ...' is this sentence needed? Why not simply giving the classical setting with integration over a non-negligible region?
* p.3: the notation $g_{p,q}$ is never defined.
* p.3: the link between Theorem 1 and the log-linear model is unclear: how the two summations become only one? The summation in the Theorem uses the number of sample $n$ while the summation in (5) depends on the dimensionality of the subset. This is particularly unclear.
* p.3: notation $\perp$ is not defined.
* p.4: the variable $Z$ is defined but never used.
* p5.: 'We can always use the natural gradient ...' this sentence is unclear.
* p.5: 'uniform distribution' on $[0,1]$? What are the 'expected number of samples' that are used?
* p.8: Table b: What are $T$ and $W$? How is it possible to have a joint process for APP-1?

---

> ### Author Response · Authors · 2020-11-24
> **Response to the comments of Reviewer #3**
>
> Thank you very much for your thorough read of our paper. Below is the response to each of your concerns.
>
> > what is the structure of the intensity function? Is it log-linear?
>
> No, in the synthetic experiments, the intensity function is generated by a mixture of Gaussians. This is why KDE with a Gaussian kernel slightly outperforms.
>
> > Does this imply that only simultaneous interactions are considered in this work, as suggested by the second sentence of Section 3.1?
>
> Yes that is correct, only simultaneous interactions are considered in this work for process 1 and 2. The multivariate function is possible using our formulation as mentioned between Eq.(6) and Eq.(7), but the space of its computation is extremely large for anything greater than 2 dimensions. So we did not investigate the multivariate function any further.
>
> >How would the method compare to others when the log-linearity is not satisfied
>
> Thank you for the suggestion. We have not studied this case in detail. If the log-linearity is not satisfied, then our graph structure is flexible enough to be able to adjust parameter $k$ so that only the parameters which satisfy the log-linearity are included in the model.
>
> >model comparison relies on a negative log-likelihood criteria which does not account for the complexity of the models. If possible, some more general criteria such as AIC should be considered
>
> Thank you for the suggestion. It is not entirely clear to us whether or not it is appropriate to use AIC for an infinite non-parametric model such as DP-beta and RKHS. Since we are using the negative test log-likelihood on not the training dataset but the test dataset, we directly examine the generalizability of each method to out-of-distribution samples. Therefore the complexity of the models is implicitly taken into account in our experiments, which is the standard approach in machine learning.
>
> > p.3: notation $\bot$ is not defined.
>
> It is now defined under the first paragraph of section 2.2 in our revised version.
>
> >p.4: the variable $Z$ is defined but never used.
>
> It is used in Eq.(9)
>
> >p.3: the link between Theorem 1 and the log-linear model is unclear: how the two summations become only one? The summation in the Theorem uses the number of samples while the summation in (5) depends on the dimensionality of the subset. This is particularly unclear.
>
> The link between Theorem 1 and the log-linear model is Eq.(5). The two summation becomes one summation is explained in between Theorem 1 and Eq.(5). The full detail of the derivation can be found in [Braun 2009].
>
> > p.5: 'uniform distribution' on $[0, 1]$? What are the 'expected number of samples' that are used?
>
> Yes this means that $T = 1$. The expected number of samples is explained in the first sentence of Section 3.1 and last sentence in the first paragraph in Section 3.2.
>
> > p.8: Table b: What are T and W? How is it possible to have a joint process for APP-1?
>
> T and W represents Tuesday and Wednesday. We are still able to estimate the joint intensity function from a first order model. For the first order model, we fix all other values of $\theta$ to be $0$. However, this does not mean that $\eta$ or $p$ are equal to $0$. Therefore an intensity function can still be estimated even if $\theta = 0$. This is more apparent in our high dimensional experiment in Fig.5 (and Fig 8 for the full result). Please see [Luo and Sugiyama 2019] for more details.

---

### Official Review · AnonReviewer1 · 2020-10-29
**Paper too unclear to understand & assess contributions**

**Rating:** 4
**Confidence:** 4

**Review:**

## Summary

This work claims to address the problem of learning the structure of interactions between multiple point processes. This can, e.g., be used to predict co-occurrences between different types of events. The authors develop a method that (a) is theoretically principled and (b) scales to large datasets.

## Reasons for score

The overall problem is clearly relevant, and some aspects of the contribution appear to be promising (connection to GAMs, approach to inference). However, in its current form, the paper is excessively unclear, to the point where I am not sure that I understand the precise problem the authors are trying to address (despite my familiarity with the area).

## Pros

- if the problem is that of understanding correlations between multiple point processes, then that is an important problem that is clearly relevant to the ML community.
- The approach builds on generalized additive models, which gives their approach a solid theoretical foundation
- The inference algorithm based on minimizing KL-divergence appears to be interesting.

## Cons

- the paper lacks clarity throughout, making it difficult to understand the problem, let alone the contribution itself (examples below).
- the experiments are not explained in sufficient details (NYC taxi dataset for example), and as such they are not supporting the contribution.

## Comments

Referring to "stochastic processes" is too general—maybe talking about "point processes" would be better. For example, a Gaussian process is a stochastic process, but doesn't have anything to do with the processes considered in this work.

Section 2, "Formulation" is confusing. Usually, a multi-dimensional / spatial poisson process is a *single* point process defined on $\mathbf{R}^D$, and a realization of that process is a collection of points in D-dimensional space. This is what, e.g., Flaxman et al. (2017) consider.

In this paper, the authors seem to assume that every dimension is a different process, and somehow events in each dimension are grouped together based on their arrival order ("Each $t_i$ is the time of occurrence for the $i$-th event across $D$ processes and $T$ is the observation duration"). I am not even sure this is always possible, since it seems to assume that every process / dimension has exactly $N$ events during the interval $[0, T]$. Thus, I fail to understand even the basic foundation on which their work builds.

It seems to me that the authors are really considering a *multivariate* Poisson process, where the intensity function of each process are dependent. In that case, the likelihood would be different from (1). Can the authors comment on this?

Experiments:

- "the sample size is randomly chosen by the mixture of Gaussians" - what does this mean? a GMM does not define a distribution over number of samples.
- where is the notion of timestamps in the synthetic GMM experiments?
- Figure 3 does not seem to be referenced in the text. How can the intensity function be 1D for if the experiments are with 2D Gaussians?
- NYC taxi dataset: how is space taken into account? Defining a "common pick-up" as one that happened within at 1 min interval but at opposite ends of the city seems pointless.

Other comments / questions:

- How does your work compare / contrast to multivariate Hawkes processes?
- In what way is Flaxman et al. 2017 non-convex?
- " ...where the different events are marked so that a particular subset of events can be used to generate the intensity function" what does this mean?

## POST-REBUTTAL

Thank you for rebuttal which has helped me, to some extent, to understand your model. I still believe the formulation needs to be clarified before the contributions can be assessed objectively.

For example, on page 2:

> Given a realization of timestamps $\mathbf{t}_1, \mathbf{t}_2, \ldots ,\mathbf{t}_N$ with $\mathbf{t}_i \in [0, T]^D$ from an inhomogeneous (multi-dimensional) Poisson process with the intensity $\lambda$. Each $\mathbf{t}_i$ is the time of occurrence for the $i$-th event across $D$ processes and $T$ is the observation duration.

This is confusing: only a single one of the dimensions will likely represent "time". Other dimensions will be space, etc. Furthermore, why would every dimension need to be within $[0, T]$? Different dimensions might use different units and be in different ranges.

I am upgrading my score slightly, because I sense that your contributions might be interesting once properly explained, but in the present state I believe the paper is not ready for publication.

---

> ### Author Response · Authors · 2020-11-24
> **Response to the comments of Reviewer #1 (1/2)**
>
> Thank you for the detailed comments on our submission. Our response to each of your concerns are below.
>
> 0\) Yes that is correct, currently our work focuses on point processes only, we have revised the wording to refer to the more specific case of point processes instead of stochastic processes.
>
> 1\) Yes, you are correct, our formulation in Section 2.1 and 2.2 have exactly N observations in each of its processes/dimensions. The exact same number of N is quite common in spatial-temporal problems, where the dimensions of the Poisson process represent different spatial dimensions and time dimensions.
>
> Our point process model is different with the traditional multivariate point process. Our point process model is decomposing a multi-dimensional point process (each point is multi-dimensional, that is, $(x,y,t)_{i=1}^{N}$, where $x$ and $y$ represents two spatial dimensions and $t$ represents the time dimension) to multiple lower-dimensional representations. For example for our first order model ($k = 1$), we have one-dimensional representation of the Poisson process, that is $(x_i)_{i=1}^{N}, (y_i)_{i=1}^{N}$ and $(t_i)_{i=1}^{N}$.
>
> For example in our taxi dataset. A single pick up event is a point in two spatial dimensions and one time dimension (example in the paragraph above). As you can see, this is a very challenging problem, because there will be very few observations (or even no observations) of an event at that precise spatial location and time.
>
> Our model provides a way to estimate the intensity function for these simultaneous events in Poisson processes. The intensity for these simultaneous events are generally very low and therefore the current approaches in literature are unable to provide a solution to this problem.
>
> The joint intensity considering the spatial dimensions and time dimensions is very rarely addressed in the current literature. However, it is a very important problem which is required in many spatial-temporal problems. This is the precise problem we are trying to address. We have revised the introduction so that the problem we are trying to solve is much more clear.
>
> We have not discussed this in detail in our paper, but our formulation of the Additive Poisson Process in Section 2.3 does not have this assumption. Our proposed model actually relaxes this assumption and does not require exactly $N$ observations in each of its processes/dimensions.
>
> Though our formulation is unconventional compared to the mainstream literature in point processes, there are clear advantages in formulating a multi-dimensional point process in this way. We hope our work will inspire more research in this direction as we have strong theoretical guarantees for our model to converge because we have linked our models to GAMs and the Kolmogorov-Arnold Representation Theorem.

---

> > ### Author Response · Authors · 2020-11-24
> > **Response to the comments of Reviewer #1 (2/2)**
> >
> > > Comment 1 in Experiments
> >
> > Thank you for picking this up. This sentence was poorly phrased. We have revised it in our newly submitted version.
> >
> > > Comment 2 in Experiments
> >
> > The notion of “time” can be represented as the location of each point in space, where the smaller value occurs before a larger value. This is similar to applying Poisson processes to spatial problems.
> >
> > > Comment 3 in Experiments
> >
> > Thank you for pointing this out. We have referred to Figure 3 in the third sentence in Section 3.1 in our revised version. We are only considering the intensity of the simultaneous interactions between the two processes. As we have mentioned in between Eq.(6) and Eq.(7) it is possible to compute the entire intensity function, but the space to compute it is extremely large for anything greater than 2 dimensions. So we did not investigate the multivariate function any further.
> >
> > > Comment 4 in Experiments
> >
> > The common pick up time is used to see how much resources are needed at a given time such as taxi drivers, cars and operating staff to be working at a given time and currently our experiment does not take into account the notation of space. But yes, as you have correctly mentioned, space can be taken into account in the experiment to find a common pick up location to see where the taxi’s should be located in the city. Space can be included into the model by adding an additional dimension to the Poisson process. This is currently our future work.
> >
> > > Comment 1 in Other comments / questions
> >
> > The intensity function in the Poisson process is deterministic. While the intensity in a Hawkes process is stochastically excited by each new arrival occurrence. In a multivariate Hawkes process, we usually learn the conditional intensity function. Differently, our proposed approach learns the joint intensity function. Learning the joint intensity function is a much more general problem and has a wide range of applications, including all the applications that the conditional multivariate Poisson process solves.
> >
> > > Comment 2 in Other comments / questions
> >
> > [Flaxman et al. 2017] is a non-convex optimization as mentioned at the bottom of page 14 of [Flaxman et al. 2017]. This can be seen more clearly in the objective function in Eq.(3.4) in [Flaxman et al. 2017], where the first term is clearly not convex. To add to this difficulty there are at least 3 hyper-parameters to tune in this model. Even when we are given the hyper-parameters of the model, if we have initialized the weights differently, we still arrive at a different solution. This can be verified by a simple experiment.

---

### Official Review · AnonReviewer2 · 2020-11-01
**This paper proposes a reasonable method to model high-order correlations. However, its experimental result is weak and the model construction is of a question to me.**

**Rating:** 3
**Confidence:** 4

**Review:**

This paper proposes a generalized additive model to learn joint intensity functions for multiple Poisson processes. The main contribution is to consider the partial order of high-order interactions when parameterizing the model.

The proposed method is a reasonable solution, but there are several points in the paper that are particularly weak:

First, the paper does not find a proper scenario to apply the additive Poisson process model. The experimental part lacks real data analysis. In Section 3.3, they only show a quantitative result regarding two Poisson processes [T, W]. It is hard to summarize that the performance gain comes from high-order modeling without explaining it in detail. For example:
i) Why are KDE and RKHS predictions inconsistent across January through March?
ii) How do you define M?
iii) How sparse is the data?

Second, jumping from section 2.1 to 2.2 is confusing, since section 2.2 seems to have nothing to do with the Poisson process. So my first question is:
i) Can you define the likelihood in a single Poisson process model in order to consider all the interactions you mentioned in section 2.2?
After carefully reading the setup in section 2.2, I feel that the entire method is based on a crude approximation of Poisson processes. For example, I’m curious about:
ii) Does your method really depend on discretizing the time window? What if you let M goes to infinity? I’m particularly interested in this question because the original Poisson process likelihood model does not require discretization.

To conclude, this paper proposes a reasonable method to model high-order correlations. However, its experimental result is weak and the model construction is of a question to me.

---

> ### Author Response · Authors · 2020-11-24
> **Response to the comments of Reviewer #2**
>
> Thank you for your interest in reviewing our paper. The response to your questions are below.
>
>
> > i) Why are KDE and RKHS predictions inconsistent across January through March?
>
> We have revised our experimental results with more consistent experimental results. Previously we have received inconsistent results for RKHS because the optimization of RKHS is nonconvex. Even if given the hyper-parameters of the model, if we have initialized the weights differently, we arrive at a different solution. For more consistency, we have run the experiment more times to achieve a much more consistent result. However, finding consistent results is difficult because it is non-convex. We have rerun the experiments with more runs and have found more consistent results.
>
> > ii) How do you define $M$?
>
> M is defined in the first paragraph in Section 2.2 to be the number of bins used for the discretization. The selection of M is really application dependent. If we have domain knowledge, it is recommended that this parameter be selected empirically or by domain experts. However, if this is not possible for the given application, then we recommend cross-validation as discussed in the first paragraph in Section 3.
>
> > iii) How sparse is the data?
>
> The first sentence in Section 3.1 shows how sparse the data is for the two dimensional process. The last sentence in the first paragraph in Section 3.2 explains how sparse the data is for the high dimensional case.
>
> > i) Can you define the likelihood in a single Poisson process model in order to consider all the interactions you mentioned in section 2.2?
>
> In our revised version we have introduced the log-likelihood with all its interactions at the end of Section 2.1.
>
> > ii) Does your method really depend on discretizing the time window? What if you let M goes to infinity?
>
> If we let M approach infinity, we arrive at the equation for GAM in Eq.(4), which is the continuous time representation (with a different normalizing constant and the function f is selected to be an exponential). However, there is no efficient technique to estimate the parameters directly in Eq.(4). Also, how we define a simultaneous event is not clear in Eq.(4). If we assume that the simultaneous events  is 1/M, when M approaches infinity, this means that there are no simultaneous events. Though it may be possible to define simultaneous events which may not be directly linked to M, this appears to be non-trivial.
>
> For the discretized model, the occurrence of simultaneous events only becomes apparent during the discretization. We define a simultaneous event to be two events which occur in the same bin. Now that we have defined what is a simultaneous event, the question is how do we efficiently estimate the parameters of our model.
>
> Our contribution is finding the relationship between Poisson processes and GAMs and to propose a solution to approximate the intensity function in Section 2.2 using the information geometric formulation of the log-linear model to formulate the discretized model. Then our optimization is explained in Section 2.3 by minimizing the KL divergence from a set of samples.

---

### Decision · Program_Chairs · 2021-01-07
**Final Decision**

**Decision:**

Reject

**Comment:**

This paper proposes a method for modeling higher-order interactions in Poisson processes. Unfortunately, the reviewers do not feel that the paper, in its current state, meets the bar for ICLR. In particular, reviewers found the descriptions unclear and the justifications lacking. While the responses did aid the reviewers understanding, the paper would benefit from rewriting and more careful thought given to the experimental design.